# Force Sensing on Cells and Tissues by Atomic Force Microscopy

**DOI:** 10.3390/s22062197

**Published:** 2022-03-11

**Authors:** Hatice Holuigue, Ewelina Lorenc, Matteo Chighizola, Carsten Schulte, Luca Varinelli, Marcello Deraco, Marcello Guaglio, Manuela Gariboldi, Alessandro Podestà

**Affiliations:** 1CIMAINA and Dipartimento di Fisica “Aldo Pontremoli”, Università degli Studi di Milano, Via Celoria 16, 20133 Milan, Italy; hatice.holuigue@unimi.it (H.H.); ewelina.lorenc@unimi.it (E.L.); matteo.chighizola@unimi.it (M.C.); carsten.schulte@unimi.it (C.S.); 2Department of Research, Fondazione IRCCS Istituto Nazionale Tumori, Via G. Venezian 1, 20133 Milan, Italy; luca.varinelli@istitutotumori.mi.it (L.V.); manuela.gariboldi@istitutotumori.mi.it (M.G.); 3Peritoneal Surface Malignancies Unit, Colorectal Surgery, Fondazione IRCCS Istituto Nazionale Tumori, Via G. Venezian 1, 20133 Milan, Italy; marcello.deraco@istitutotumori.mi.it (M.D.); marcello.guaglio@istitutotumori.mi.it (M.G.)

**Keywords:** Atomic Force Microscopy, colloidal probe, biosensors, glycocalyx, extracellular matrix, mechanobiology

## Abstract

Biosensors are aimed at detecting tiny physical and chemical stimuli in biological systems. Physical forces are ubiquitous, being implied in all cellular processes, including cell adhesion, migration, and differentiation. Given the strong interplay between cells and their microenvironment, the extracellular matrix (ECM) and the structural and mechanical properties of the ECM play an important role in the transmission of external stimuli to single cells within the tissue. Vice versa, cells themselves also use self-generated forces to probe the biophysical properties of the ECM. ECM mechanics influence cell fate, regulate tissue development, and show peculiar features in health and disease conditions of living organisms. Force sensing in biological systems is therefore crucial to dissecting and understanding complex biological processes, such as mechanotransduction. Atomic Force Microscopy (AFM), which can both sense and apply forces at the nanoscale, with sub-nanonewton sensitivity, represents an enabling technology and a crucial experimental tool in biophysics and mechanobiology. In this work, we report on the application of AFM to the study of biomechanical fingerprints of different components of biological systems, such as the ECM, the whole cell, and cellular components, such as the nucleus, lamellipodia and the glycocalyx. We show that physical observables such as the (spatially resolved) Young’s Modulus (YM) of elasticity of ECMs or cells, and the effective thickness and stiffness of the glycocalyx, can be quantitatively characterized by AFM. Their modification can be correlated to changes in the microenvironment, physio-pathological conditions, or gene regulation.

## 1. Introduction

In recent years, there has been a growing interest in studying the physical properties of biological samples, such as tissues, single cells, and their microenvironment, to better understand how they change during the progression of diseases, such as cancer [1,2,3], and how they influence each other in their mutual interaction [4,5,6,7,8,9,10,11]. The extracellular matrix (ECM), which is a fundamental component of the cell microenvironment, is a ubiquitous acellulated component present in all tissues, comprising molecules that are secreted by cells and assembled to form specific insoluble components; the ECM plays a fundamental role as a scaffold for cell growth, in the regulation of cell-cell and cell-matrix signaling, also affecting cell mechanics mainly through the remodeling of the cytoskeleton, and determining cell fate [4,5,12,13,14,15,16]. There is a reciprocal interaction between the ECM and the cells, allowing the active modification of the ECM structure and composition, which affects its mechanical properties as well [9,17,18,19]. An important physical layer is located between the cell membrane and the cell microenvironment: the glycocalyx, also known as the pericellular matrix. The glycocalyx is a surface brush layer that is present on every cell, and made of glycoproteins, proteoglycans and polysaccharides [16,20]. As the first contact interface between the cell and its microenvironment, the glycocalyx plays an important role in their mutual interactions [21]. The glycocalyx acts as a water reservoir, helps in the transport of metabolites and control of the signaling molecules [20,22], regulates integrin clustering and focal adhesion maturation [21,23,24,25]. The characterization of the glycocalyx thickness in relation to different pathological states of the cell would help to understand the communication between cells and the ECM [26]. More importantly, a link has been demonstrated between glycocalyx and cancer: tumoral cells tend to show a wider distribution of glycocalyx chain lengths compared to normal ones [27]; moreover, their bulky composition seems to favor the metastatic spread [28,29,30]. The physical characterization of glycocalyx has a good potential in cancer research, both as a cancer biomarker [20,25,27,30,31] and as a therapeutic target, since the reduction or degradation of the glycocalyx has been reported to reduce cell migration and suppress cell growth [32,33].

The study of the mechanical properties of cells and tissues in the context of health and disease implies the need for reliable instrumentation and methods. Atomic Force Microscopy (AFM), which is able to both sense and apply forces at the nanoscale, with sub-nanonewton sensitivity, represents an enabling technology and a crucial experimental tool in biophysics and biomechanics [34,35,36,37].

In AFM, an elastic cantilever with an intrinsic spring constant k in the range 0.05–50 N/m is used as both a force sensor and a force transducer (Figure 1A). The surface force F is applied on the cantilever, typically at its end, where a micro-tip with a radius of curvature of typically 2–100 nm is located. The force induces a vertical cantilever deflection $z\;=\;F/k_{\mathrm{eff}}$, which is typically measured by an optical beam deflection apparatus [38,39,40] (Figure 1), or in some cases by an interferometer [41]. In the above equation, an effective spring constant $k_{\mathrm{eff}}$ is used, rather than the intrinsic one, to account for specific features of the loading configuration, such as the cantilever mounting angle θ (usually θ = 10°–15°), the tip height, the loading point position, etc. [42]. In the simplest case of a negligibly small tip at the very end of the cantilever, $k_{\mathrm{eff}}=k/\cos^{2}\left(\theta \right)$.

A small spring constant provides high force sensitivity, meaning that a small force produces a large, easily measurable, deflection. A lower limit to the measurable deflection (and therefore to the measurable force) is set by the thermal noise z_th_ of the cantilever, which can be estimated from the equipartition theorem as $z_{\mathrm{th}}=\sqrt{k_{B}T/k}$, with k_B_ and T being the Boltzmann constant and the absolute temperature, respectively [43,44]. The minimum, thermal noise, limited detectable force that can be measured dynamically with an instrumental bandwidth BW is $F_{\mathrm{th},\min}=\sqrt{4k_{B}\mathrm{TbBW}}$, where b is the damping coefficient (the proportionality factor between the tip velocity and the viscous force). Equivalently, since $b=k/\left(2{\pi f}_{R}Q\right),$ $F_{\mathrm{th},\min}=\sqrt{2k_{B}\mathrm{TkBW}/\left({\pi f}_{R}Q\right)}$; Q and f_R_ being the quality factor and the resonance frequency of the cantilever, respectively; similar expressions for the minimum force gradients can be obtained [45,46].

Besides measuring the tip-sample interaction force with sub-nN sensitivity, AFM allows us to reconstruct the tip-sample distance corresponding to the force measurements, which translates into a sample deformation after contact is established [47]. The possibility of measuring force vs. distance, spatially resolved with nm resolution, assigns AFM a leading position as a force (bio) sensing technique. AFM is at present an enabling technology and a crucial experimental tool in biophysics and biomechanics, allowing both force spectroscopy and nanomechanical characterization of biologically relevant interfaces and systems.

In this work, we present an overview of how AFM can be used as an enabling force-sensing technology for the study of biological systems at different spatial and force scales. To this purpose, we report on the application of AFM to the study of biomechanical fingerprints of several components of biological systems, such as the ECM, the whole cell, and cellular components, such as the nucleus, lamellipodia and the glycocalyx. We show that physical observables such as the (spatially resolved) Young’s Modulus of elasticity of ECMs or cells and the effective thickness and stiffness of the glycocalyx can be quantitatively characterized by AFM. In particular, we carried out three representative experiments from the microscale to the nanoscale: mechanics on healthy and neoplastic decellularized tissue from one patient with peritoneal metastasis; mechanics of three bladder cancer cell lines who are representative of the progression of urothelial bladder cancer; and eventually glycocalyx characterization of those cell lines. We demonstrate through these illustrative experiments the high sensitivity that can be achieved with AFM on detecting small changes in the biomechanical properties of biological samples, from single living cells and their biomolecular components to tissues; the modification of these biophysical observables can in turn be correlated to changes of the cell microenvironment, physio-pathological conditions of the tissue and related organs, or gene regulation phenomena.

## 2. Materials and Methods

### 2.1. Sample Preparation

#### 2.1.1. Human Tissues

Peritoneal tissue was collected from one patient with peritoneal metastatic colorectal carcinoma (CRCPM) who underwent surgical resection at the Peritoneal Malignancies Unit of the IRCCS Foundation, Istituto Nazionale dei Tumori di Milano, Milan, Italy. The patient was staged according to the World Health Organization (WHO) classification. The study was approved by the Institutional review board (249/19) and was conducted in accordance with the Declaration of Helsinki, 2009. Written informed consent was acquired.

Omentum-derived CRCPM lesion and apparently normal tissue (>10 cm from the metastatic lesion) were harvested. Tissues were frozen in liquid nitrogen and used to develop the decellularized ECMs.

#### 2.1.2. Decellularized Extracellular Matrices

Decellularized extracellular matrices were obtained from the omentum fold of human peritoneum from a patient with metastases derived from colorectal cancer. The decellularization was performed as in Genovese et al. [48]. The success of the decellularization procedure was already verified in the work from Varinelli et al. [49]. The ECM samples were embedded in optical cutting compound (OCT) and frozen in 2-propanol, then kept in a liquid nitrogen bath.

ECM slices of approximately 100 µm thickness were cut with a microtome (Leica) and attached to positively charged poly-L-lysine-coated glass coverslips (ThermoFisher Scientific, Waltham, MA, USA) exploiting the electrostatic interactions to improve the attachment. The samples were stored at −4 °C until AFM analysis.

#### 2.1.3. Cells

Three commercial human bladder cancer cell lines of different grades (a marker of invasiveness), kindly provided by Dr. M. Alfano (San Raffaele Hospital, Milano), were used (see Table 1) [50,51]. The cell lines were cultured in RPMI medium containing 2 mM L-glutamine supplemented with 10% FBS, 1% penicillin/streptomycin, and 1% amphotericin and grown in an incubator at 37 °C and 5% CO_2_ (Galaxy S, RS Biotech). All reagents and material were from Sigma-Aldrich (St. Louis, MO, USA) if not stated otherwise.

For AFM measurements, the cells were plated the day before on glass bottom Petri dishes (∅ 40 mm Willco Wells) coated with poly-L-lysine (0.1% *w*/*v* for 30 min at RT) to improve cell attachment, in the RPMI medium without phenol red, as it can cause damage to the AFM probe holder.

### 2.2. Histochemistry

Before histochemistry staining, ECM Formalin-Fixed Paraffin-Embedded (FFPE) sections were cut into slices and dewaxed in xylene, rehydrated through decreasing concentrations of ethanol, and washed with water. Slices were stained with van Gieson trichrome (Bio-Optica, Milan, Italy) following the manufacturers’ instructions.

### 2.3. Force Sensing with the AFM

All the experiments have been performed using a Bioscope Catalyst AFM (Bruker) mounted on top of an inverted microscope optical microscope (Olympus X71, Tokyo, Japan). The system was isolated from noise using an active antivibration base (DVIA-T45, Daeil Systems, Wonsam-myeon, South Korea) placed inside an acoustic enclosure (Schaefer, Vigilliano Biellese, Italy). Living cells were measured using a thermostatic fluid cell, with the temperature of the medium kept at 37 °C by a temperature controller (Lakeshore 331, OH, USA). The measurements on ECMs were performed at room temperature in a droplet of PBS confined on the glass slide using a hydrophobic pen.

Homemade colloidal probes were produced by attaching borosilicate glass or soda lime spheres to rectangular tipless cantilevers (NanoandMore TL-FM and MikroMasch HQ:CSC38/Tipless/No Al); both production of the probes and characterization of their radius were performed according to custom procedures [52]. Different sphere radii R and spring constants k of the probes were selected according to the needs of each experiment (Table 2).

The raw output on an AFM force measurement consists of the raw deflection signal ∆V of the cantilever, measured by the optical beam deflection (OBD) system in volts, as a function of the z-piezo displacement d_p_, in nm units (Figure 1A,B). Depending on the AFM system, the z-piezo can displace either the probe or the sample.

Exploiting two calibration parameters, the effective spring constant k_eff_ (N/m) and the deflection sensitivity S (or inverse optical lever sensitivity invOLS, in units of nm/V), the raw deflection ∆V can be transformed into the cantilever deflection z, in nm, and the latter deflection can then be transformed into a force, in nN:z = S ∆V(1)
F = k_eff_ z = k_eff_ S ∆V(2)

The tip-sample distance d can be calculated as:d = d_p_ + z − d_0_(3)

In Equation (3), d_p_ decreases as the tip gets closer to the sample surface, and z is positive when the cantilever is deflected upwards, under the action of a repulsive force, and negative in the opposite case. The parameter d_0_ represents the location along the d_p_ + z axis where the tip-sample distance is zero. The identification of d_0_ is easy when the tip is ramped against a stiff substrate, since all data points belonging to the contact region of the force vs. d_p_ + z distance curve must collapse along a vertical line, whose corresponding mean abscissa value is d_0_. On deformable surfaces, d_0_ is typically obtained through a fit of a suitable contact mechanics model (typically the Hertz model [53,54,55]) to the F vs. d_p_ + z curve [56].

The cantilever spring constant has been calibrated using the thermal noise method [43,44], and fine corrections were applied to account for geometrical and dimensional issues [42,57]. The deflection sensitivity S of these probes was calculated according to different procedures: either as the inverse of the slope of the raw deflection ΔV vs. z-piezo displacement d_p_ curve (Figure 1B) acquired on a stiff substrate [47], or via the SNAP method [58], assuming a previously accurately calibrated intrinsic spring constant as reference.

After identification of d_0_ and proper translation of the distance axis, negative distances correspond to deformations, i.e., indentations of the deformable sample. In nanomechanical tests, the negative semiaxis is the relevant one, and an indentation axis δ can be defined as: δ = −d, for d < 0.

Processing of the data was carried out using custom routines written in a Matlab environment (Mathworks, Natick, MA, USA).

The precise alignment of AFM and optical images was possible using the Bruker MIRO software and allowed us to choose the regions of interest for ECMs and cells. For the ECMs, the regions for measurements were chosen based on the evaluation of optical images; thanks to the reduced thickness of the slices and their consequent transparency, it was possible to select regions with moderate roughness and better structural integrity.

#### Indentation of Living Cells and ECMs by AFM

Cells

For the mechanical characterization of cells, the Hertz model was applied [53,54,55]. To extract the value of the Young’s Modulus (YM) E, which is the proportionality constant (within the limits of linear elastic response) of stress σ (force per unit area, in Pa) and strain ε (relative deformation): E = σ/ε. The YM is an intrinsic elastic property of a material and provides a measure of sample rigidity. According to the Hertz model for a parabolic indenter, the force vs. indentation relation is:(4)$$F\;=\frac{4}{3}\frac{E}{1-\nu^{2}}R^{\frac{1}{2}}\delta^{\frac{3}{2}}$$
which is accurate as long as the indentation δ is small compared to the radius R. In Equation (4), ν is the Poisson’s coefficient, which is typically assumed to be equal to 0.5 for incompressible materials.

When indenting compliant thin cells (typically a few microns tall at their maximum height, i.e., above their nucleus), the finite-thickness effect must be taken into account. This effect is related to the influence of the stiff glass substrate underneath the cells, which confines the strain and stress fields and makes the elastic cell response stiffer, i.e., the measured Young’s Modulus larger [55,59,60,61,62]. The finite-thickness correction depends on the ratio χ of the contact radius $a\;=\sqrt{R\delta }$ to the sample thickness h (and not trivially on the ratio δ/h):(5)$$\chi \;=\frac{\sqrt{R\delta }}{h}$$

Notably, AFM provides the unique capability of measuring simultaneously both the height and elastic properties of a sample (combining topographic and mechanical imaging [55]), therefore allowing us to implement point by point corrections that depend on ratios like the one reported in Equation (5) as for the present work.

A polynomial correction factor ∆(χ) can be applied to the Hertz equation (Equation (4)), under the hypothesis that cells are partially bound to the substrate, and this allows us to extract correct YM values irrespective of the local thickness of the sample. Following Dimitriadis et al.’s work [55,59]:∆(χ) = 1+1.009χ + 1.032χ^2^ + 0.578χ^3^ + 0.051χ^4^(6)

Introducing the rescaled force F′(δ) = F(δ)/∆(χ(δ)), Equation (4) can be replaced by the formally similar Equation (7):(7)$$F^{\prime }\;=\frac{4}{3}\frac{E}{1-\nu^{2}}R^{\frac{1}{2}}\delta^{\frac{3}{2}}$$

For the evaluation of the Young’s Modulus of single and clustered cells, at least 10 cells, each from three to five Petri dishes, were measured. For each measurement, FCs on both the substrate and the cells were acquired with a minimum of 10 FCs on the surrounding substrate and 100 FCs on the cell; this allowed us to calculate the local height of each single cell [55]. For full mapping of single cells/clusters, FCs were collected on a grid spanning an area of up to 100 μm × 100 μm around the cells, including both cells and substrate. Each FC contained 8192 points, with ramp length l = 15 μm, maximum load Fmax = 5–10 nN, ramp frequency f = 1 Hz. The probe radius was R = 5.7 μm or R = 6.4 μm. Typically, indentation up to 2 μm was achieved.

We created masks based on the obtained topographic maps to select force curves belonging to distinct regions: the nuclear region and its complement, which is the union of cell perinuclear and peripheral regions [55,62].

The same data were used for both mechanical analysis of cells and glycocalyx characterization. The first 10% of the indentation range after the contact point is usually attributed to the contribution of the glycocalyx [21,27,31,63,64]. The YM of the cells was extracted by fitting the Hertz model to the FCs in a suitable sub-interval of the remaining 10–90% indentation, typically identified as the range where the value of E does not change with indentation (Figure 2).

Glycocalyx thickness

The glycocalyx characterization was performed following the soft brush model implemented by Sokolov et al. [27,31,65,66]. The separation H between the tip and the cell membrane can be expressed as (Figure 3A):H = d_p_ − d_0_ + δ + z(8)
where d_0_ is the position of the non-deformed cell membrane (the contact point referred to Hertzian indentation), d_p_ is the relative z-piezo position, and δ and z are indentation and cantilever deflection, respectively, as previously defined.

The indentation δ in Equation (8) is calculated using the standard Hertz model (Equation (4)). When the glycocalyx is completely compressed (which typically occurs well before cell indentation is significant), H is negligibly small; it follows that if one plots the force as a function of H, the force points related to the Hertzian indentation of the cell collapse along a vertical line at H **≅** 0. This can be seen in Figure 3B, where the force for H > 0 can be identified with F_glycocalyx_, the force exerted by the glycocalyx. The latter force can be modeled as [65]:F_glycocalyx_ = 100 k_B_T R N^3/2^ L exp(−2πH/L)(9)
where L and N are the effective thickness and the grafting density of the pericellular brush, respectively.

ECM

The mechanical properties of ECMs were studied by collecting sets of typically 15 × 15 force curves (force volumes, FV) in different macroscopically separated regions of the sample. Each selected region was typically as large as 115 μm × 115 μm. Each FC contained 8192 points, with ramp length L = 15 μm, maximum load Fmax = 800–1500 nN, ramp frequency f = 1 Hz and R = 12.8 μm. Typical maximum indentation was 5–9 μm. For each patient condition, 2000–5000 FCs were obtained.

The value of the YM of elasticity of ECM was extracted as described previously for cells. The Hertz model was fitted to the [20–80%] indentation range of the FCs (Figure 2) without the finite-thickness correction (given the large thickness of the slices, χ ≤ 0.1). On tissues and ECMs, the first 20% of the FCs is typically neglected, due to the contribution of superficial non-crosslinked fibers, surface roughness -related issues, etc. [5].

### 2.4. Statistics

For both cells and ECM, the mean median value Ē_med_ of the Young’s Modulus E (or the mean values of other observables) has been evaluated for each tested condition, averaging over cells or measured ECM samples. The associated errors were calculated adding in quadrature to the standard deviation of the mean $\sigma_{{\bar{E}}_{med}}$ an instrumental error of 3%, calculated through a Monte Carlo simulation, as described in [55,67], based on the uncertainties in the calibration parameters (5% for the deflection sensitivity S, 10% for the spring constant k).

The assessment of the statistical significance of the differences among the tested conditions was carried out using the two-tailed *t*-test. A *p*-value < 0.05 was considered statistically significant.

For the glycocalyx analysis, the length L of the glycocalyx for each force curve located on the nucleus was extracted, by fitting Equation (9) to the data, and the histograms of the logarithmic values were reported. Median values were calculated.

## 3. Results

### 3.1. AFM at the Microscale: Mechanical Properties of ECMs

In this experiment, we carried out mechanical measurements on healthy and neoplastic decellularized extracellular matrices coming from the same patient affected by CRCPM. We measured YM values of the samples at deep indentation with a focus on their distribution.

The production of custom colloidal probes allows us to tune both the spring constant k of the cantilever and the sphere radius R, to match the typical length scale of tissues and ECMs, which is approximately 10–50 µm (cf. Material and Methods)

Exploiting the large colloidal probe radius allows us to effectively average local nanoscale heterogeneities due to the fine structure of the ECM, while capturing the overall mesoscopic mechanical response of the sample. To this purpose, it is important to achieve reasonably large (in absolute terms) indentations (5–9 µm, compared to the 100–200 µm thickness of the samples). In these operative conditions, finite-thickness effects are negligible, and we are confident to test the bulk sample properties, as in a 3D structure, and not only those of a surface layer, which in similar samples can be different from the bulk. The measured mechanical response therefore reflects the collective contribution of all components of the ECM, organized in micrometer-sized structural and functional domains [5,17,68,69,70,71,72]. Small colloidal probes, and to a larger extent sharp pyramidal tips, would permit a greater spatial resolution, but the mechanical output would be more scattered and less representative of the overall properties of the ECM [55].

In Figure 4A, we show the distribution of the logarithmic YM values from each FC taken on ECM samples. The fact that log YM values are approximately normally distributed suggests that the distribution of YM is lognormal, as it is typically observed [73].

During cancer progression, the neoplastic ECM becomes stiffer; indeed, the logarithmic YM distribution appears rigidly shifted to higher values (higher median value), while the logarithmic standard deviation is approximately preserved among normal and neoplastic conditions (Figure 4A). In Figure 4B, we show the distribution of the median YM values measured in different locations (FV) and slices of ECM are reported. Neoplastic-derived samples showed a significant increase in stiffness, and this result was in line with data already published [1,2,9,74,75,76]; this stiffening during cancer progression is related to an outcome of the tumor microenvironment remodeling and changes in the ECM composition and structure, including aggregation and realignment of ECM components, mediated by tumoral cells [9]. Reorganization of the matrix through cancer progression is also a parameter that can be seen optically, with classical staining of the ECM (Figure 5). The neoplastic ECM shows collagen accumulation (Figure 5B), while the healthy counterpart shows organized collagen fibers (Figure 5A).

### 3.2. Mechanics of Cell, down to Cellular Components

Using AFM, it is also possible to sense small mechanical changes in single cells related to different physio-pathological conditions; using colloidal probes, the spatial resolution can be good enough to discriminate among single cell components, such as nuclear and perinuclear regions and lamellipodia [55], while sharp tips allow us to discriminate fine cellular structures as small as single actin fibers [77].

We performed AFM nanoindentation on three bladder cancer-derived cell lines, RT4, RT112, and T24, with different degrees of invasiveness (Table 1), and compared their median YM values.

From the force vs. distance curves, we reconstructed three-dimensional cell morphologies and the mechanical maps, as described in [55]. All FCs, and consequently all maps, have been corrected for the contribution of the finite-thickness of the sample, as explained in the Methods.

Mapping both topography and YM and comparing with the optical image (Figure 6) allowed us to decouple the contributions of the nuclear regions and the other regions of the cells in the same cluster. Examples of the distributions and median values of the YM extracted from different parts of the cell body (e.g., nuclear vs. perinuclear and peripheral regions, or lamellipodia) are shown in Figure 7 and Figure 8.

As shown in Figure 8, the higher degree of invasiveness of cells (from RT4 to T24) correlates to a decrease in the YM (whole-cell value); this is consistent with previously published data [78]. The reported differences in YM between cell lines RT4 and RT112 and between RT4 and T24 were found to be significant, while this was not the case between RT112 and T24.

We observed a wider distribution of the YM values in the perinuclear and peripheral regions of the cells, compared to the nuclear region (Figure 7). In the perinuclear and peripheral regions, both softer and stiffer areas coexist, as shown in Figure 6C; the higher Young’s Modulus values are found at the cell-cell boundaries, where adherent junctions are present. The nuclear region exhibits a narrower distribution of YM values (Figure 7) and is stiffer than the perinuclear region, as reported previously [19,79].

It is well known that during embryonic and cancer development, cells exhibit a softening that can favor extravasating through the blood capillaries, allowing the attachment to a secondary site, favoring the metastatic spread in cancer [80]. The softening of the neoplastic cells has already been reported in breast and bladder cancer models [80,81,82,83]. It was reported that RT112 cells possess both mesenchymal and epithelial phenotypes as they are an intermediate cell line for those states [11,84]. In our case, this correlates with the intermediate YM values observed for these cells compared to RT4 and T24.

### 3.3. Down to the Nanoscale: Characterization of the Glycocalyx

As previously described, AFM is capable of sensing the mechanical resistance to compression of tissue components such as cells and ECM; it is possible to go further down along the size and force scales, characterizing even smaller and more delicate structures such as the pericellular matrix, a sugar-rich coat, called glycocalyx.

Many models have been developed for the data analysis and the characterization of the glycocalyx and similar brushes [27,85]. Here, we followed the protocol developed by Sokolov et al. [21,27,31,66], which is based on decoupling the deformation of the ultrasoft glycocalyx layer on top of the soft cell, within the acquired FCs. We applied this model to the FCs collected for the nanomechanical measurement of bladder cancer cells, to prove the feasibility of extracting more information from the same data set (see Section 2).

To better appreciate the subtle differences between the three cell lines RT4, RT112, and T24, we considered the distribution of the glycocalyx thickness values extracted from the single FCs in log scale (Figure 9). There are differences in the distributions of L values in the three cases. Compared to the intermediate grade of invasiveness (RT112), the less and most invasive cells (RT4 and T24, respectively) possess a broader distribution of brush lengths, with tails extending towards longer and shorter lengths, respectively; the median values of the glycocalyx thickness therefore tends to decrease going toward a higher degree of invasiveness, from L = 730 nm (RT4 and RT112) to L = 652 nm (T24). Nevertheless, the distribution seems to possess different modes (highlighted by the dotted vertical lines in Figure 9), and one can see that the relative importance of higher-thickness modes increases for more invasive cells, which are also characterized by the more asymmetric brush length distribution, as it was observed for tumoral cells [25,30]. These data suggest that beside the mere thickness/length of the brush, also the change of other glycocalyx physical properties such as the stiffness, the effective graft density, and degree of crosslinking should be quantitatively characterized since they are likely correlated to the transformation of a tissue from a normal to neoplastic condition.

## 4. Conclusions

In this paper, we discussed the capability of the AFM as a reliable force sensor for the study of biological systems. The representative results presented here demonstrate the possibility of using AFM nanomechanical measurements to characterize physical modifications related to specific physio-pathological conditions of cells and tissues.

As demonstrated in this work, as well as in many others [36,86,87,88,89], AFM can be used to test biological samples at several different scales, in terms of dimensions and forces, from large, rough, and relatively stiff ECMs and tissues, passing through smaller and soft cells, to extremely compliant pericellular brushes. Here, in particular, we reported on the capabilities of the instrument to characterize the mechanical differences of urothelium bladder cancer cells and the different organization of their glycocalyx brushes using the same mechanical data sets. At a larger scale, the stiffening of human-derived ECM during the progression of colorectal carcinoma could be detected.

The crosstalk between cells and their microenvironment is complex and challenging to quantitatively assess; the reliability of AFM also stands in its flexibility of measurements, which is demonstrated by the capability of AFM to both sense and apply forces in aqueous physiological conditions with controlled temperature, as well as by the possibility of resolving the measurements not only spatially, but also in the time and frequency domains; this makes the tracking by AFM of dynamic biological processes possible, including monitoring cell mechanical changes through cytoskeleton rearrangement, for example, due to drug actions or genetic modifications [77,90].

AFM and AFM-inspired instruments will likely play an increasingly important role in establishing experimental approaches for the mechanical phenotyping of cells and tissues in health and disease conditions, with the potential to develop effective early diagnostic tools based on biomechanical measurements.

## Figures and Tables

**Figure 1 sensors-22-02197-f001:**
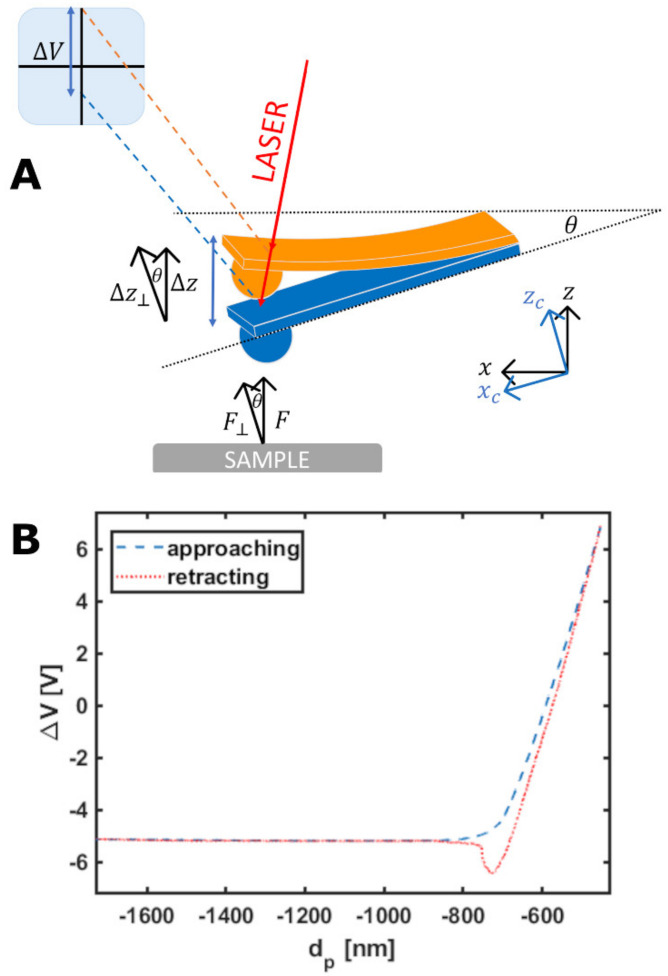
(**A**) Scheme of the optical beam deflection (OBD) system. The vertical displacement of the cantilever induced by the sensing of a force F perpendicular to the sample surface is detected on a segmented photodiode as a raw voltage signal ΔV. The cantilever is typically mounted at an angle θ with respect to the sample surface. (**B**) A raw force curve, representing the photodiode output ΔV as a function of the z-piezo displacement $d_{p}$. Both the approaching and retracting branches of the curve are shown.

**Figure 2 sensors-22-02197-f002:**
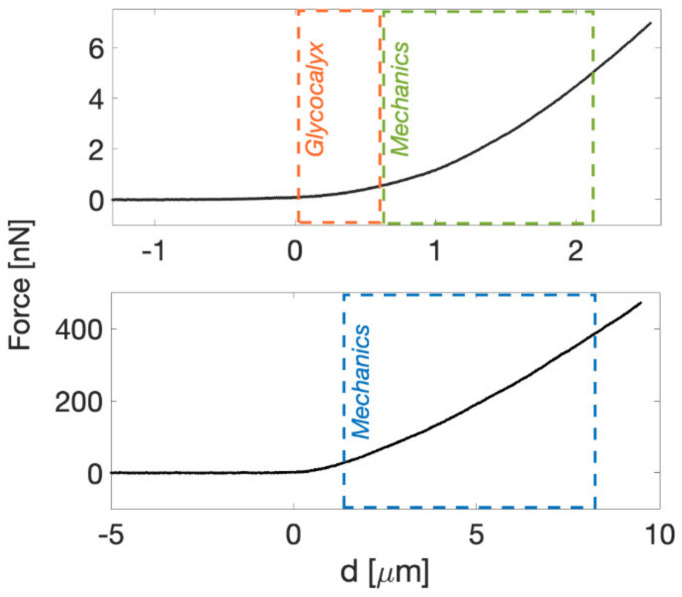
Typical approaching force curve on top of a cell (**top**) and ECM (**bottom**). The indentation range for model fitting is highlighted; on cells, we used typically [0–10%] for the glycocalyx and [10–80%] for the YM, while on ECMs we used [20–80%] for the Young’s Modulus.

**Figure 3 sensors-22-02197-f003:**
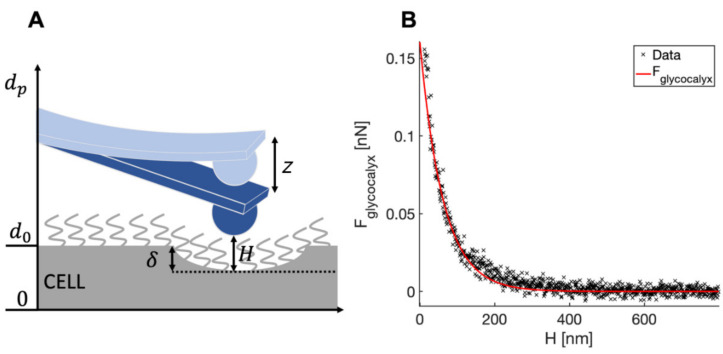
(**A**) Schematics of the distances used to determine the tip-cell membrane distance H (Equation (8)). (**B**) A typical force curve showing the force exerted by the glycocalyx as a function of the tip-cell membrane distance (the red continuous curve is the fit by Equation (9)).

**Figure 4 sensors-22-02197-f004:**
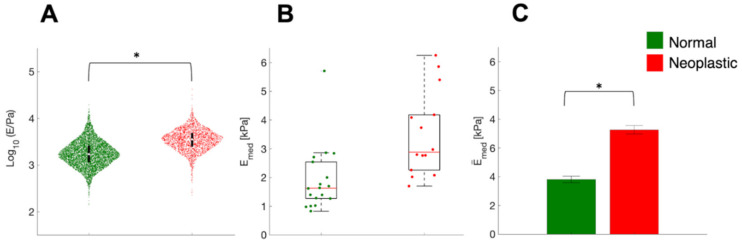
The stiffening of ECM in CRCPM samples. (**A**) Logarithmic values of the YM and their distribution in normal and neoplastic ECM samples obtained from one patient. Violin plots were plotted collecting YM values from all single FCs. Violin plots suggest that the distribution of local YM values is approximately lognormal. The circle and the black bars represent the median and the interval between 25th and 75th percentiles. (**B**) Comparison of median YM values E_med_ from each force volume in linear scale for normal and neoplastic samples from one patient. The red line represents the median value, the box encloses the interval between 25th and 75th percentiles of the sample. Whiskers go from the upper and lower limits of the interquartile range to the furthest observations, within 1.5× the interquartile range; data points beyond this limit are considered outliers. (**C**) Comparison of the mean median YM values for the two conditions tested. In (A, C), * means *p* < 0.05.

**Figure 5 sensors-22-02197-f005:**
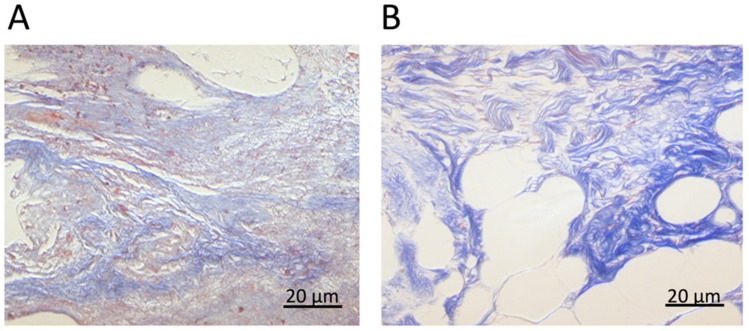
Collagen of ECM of healthy (**A**) and tumoral region (**B**) stained with van Gieson trichrome.

**Figure 6 sensors-22-02197-f006:**
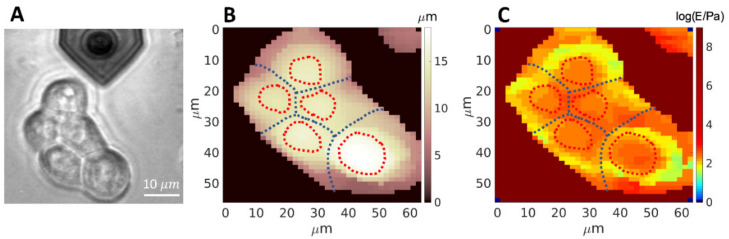
Representative images for the combined topographical and mechanical analysis of cell clusters. Optical image of a cell cluster from the RT112 cell line (**A**); topographic map (**B**), and Young’s modulus map (**C**) in logarithmic scale, of the same cluster shown in (**A**).

**Figure 7 sensors-22-02197-f007:**
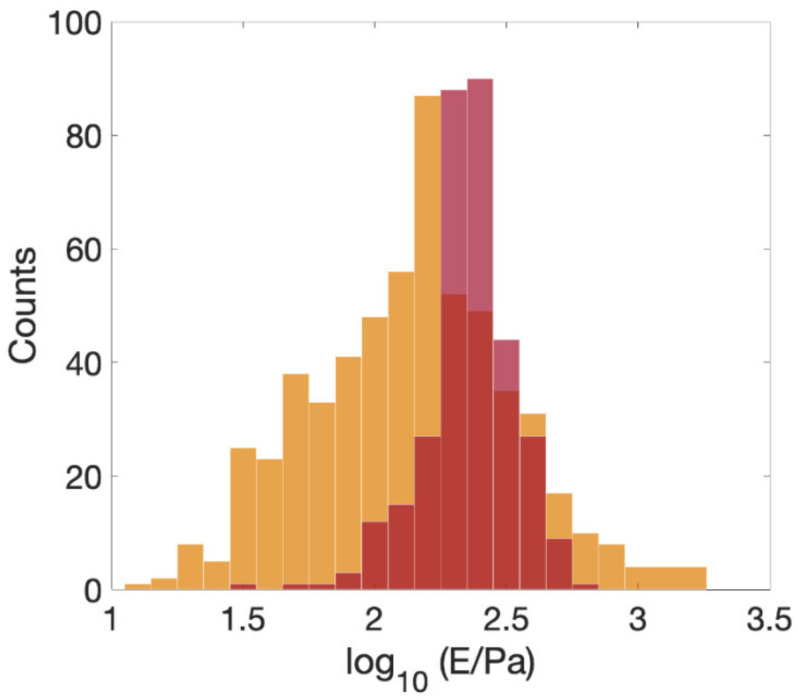
Histograms of the YM from the perinuclear and peripheral region of an RT112 cell (orange), and from the nuclear region (red).

**Figure 8 sensors-22-02197-f008:**
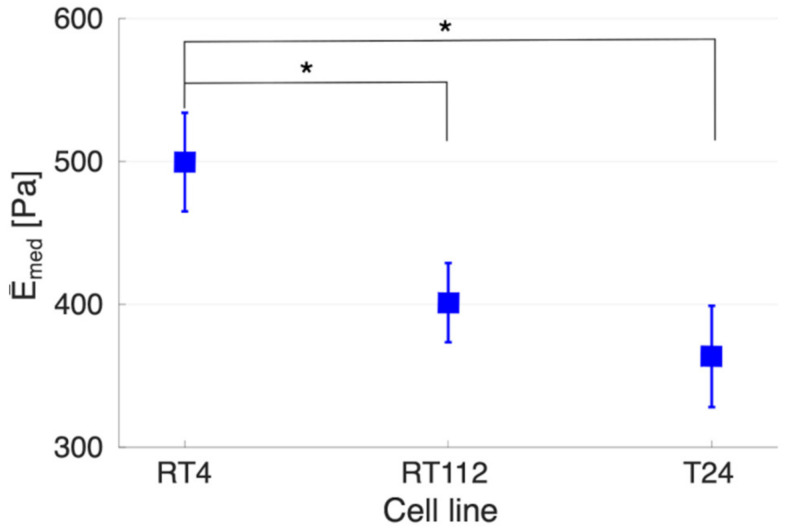
The Young’s modulus (whole cell) measured by AFM for bladder cancer cells RT4, RT112, and T24, with increasing grades of invasiveness (left to right). * Means *p* < 0.05.

**Figure 9 sensors-22-02197-f009:**
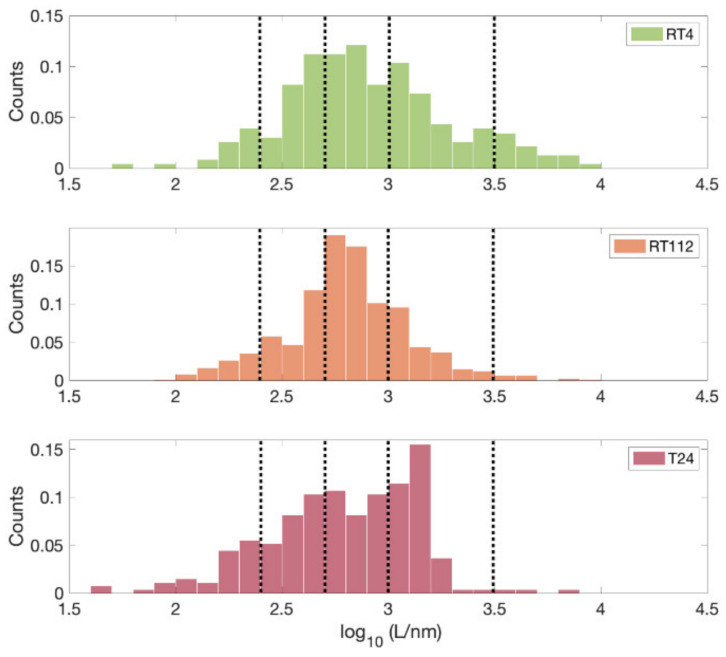
Distribution of the lengths of the glycocalyx brush (extracted according to Equation (9) from single FCs) for the three cell lines RT4, RT112, and T24. Vertical dotted lines are a guide for the eye in the tentative identification of the main modes of the distributions.

**Table 1 sensors-22-02197-t001:** Characteristics of the bladder cancer cell lines used in this work.

Cell Line	Specie/Organ	Morphology	Tumor
RT4	Human Bladder	Epithelial	Papilloma, transitional cell (Grade I)
RT112	Human Bladder	Epithelial	Papilloma, transitional cell (Grade II)
T24	Human Bladder	Epithelial	Carcinoma, transitional cell (Grade III)

**Table 2 sensors-22-02197-t002:** Radius and typical spring constant of the AFM probes used for every experiment.

Experiment	Colloidal Probe Radius (μm)	Spring Constant (N/m)
Mechanics of the ECM	20	5
Mechanics of cells	5	0.01
Glycocalyx characterization	5	0.01

## Data Availability

The data presented in this study are available on reasonable request from the corresponding author.

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
