# Peer review of "Force Sensing on Cells and Tissues by Atomic Force Microscopy"

_sensors, 2022, doi:10.3390/s22062197_

Round 1

Reviewer 1 Report

The authors used Atomic Force Microscopy (AFM) to examine the material properties of various biological samples, including decellularized ECM, cell stiffness (perinuclear and nucleus) and glycocalyx. The authors have a good understanding of the AFM principles. However, I do not support the publication of this work because the manuscript does not advance the field. All the biological systems used in the manuscript are routinely used by many research groups.

Major points:

  • The author demonstrated that AFM can be used to measure ECM stiffness. This is not a novel application of AFM because this technology has been routinely used to measure tissue ECM stiffness (e.g. ECM from brain, pancreatic, breast tumor or aorta).
  • The author claimed that the neoplastic ECM in CRCPM is stiffer than the normal tissue. However, such data is acquired from a single CRCPM patient. More patient samples (n) are needed to draw a conclusion. In addition, I would suggest the authors include some tissue staining (collagen or propidium iodide staining) to demonstrate where they acquire their stiffness map.
  • The authors demonstrated that AFM can be used to measure cellular mechanics of different compartments, such as nucleus and cytoplasm and cell surface glycocalyx. These applications have been used by many research groups, so there is not novelty. I also suggest the author to carefully examine the invasiveness level of cells they used in the paper. At the minimal, they should demonstrate that the cell lines they used (RT4, RT112, and T24) are indeed differentially invasive using standard assay such as matrigel invasion assay.
  • The author used AFM to demonstrate that the glycocalyx thickness is different between cell lines. I suggest the authors to include a control (e.g. enzymatic degradation of the glycocalyx using hyaluronidase) to ensure that they can faithfully detecting glycocalyx.

Reviewer 2 Report

This is a technically sound study and scholarly very well presented, with a topics worth bringing it to the readership of sensors. The mauscript can be published as it is.

Reviewer 3 Report

I have carefully examined the manuscript entitled Force sensing on cells and tissues by atomic force microscopy. Presented manuscript falls within the scope of the Sensors journal and present some interesting data and approach. However, I have a couple of questions and suggestions how to improve the manuscript. Please see the following comments.

My main concern is related to the novelty of the study. Authors frequently refer to the previosly published study on very similar issues as presented in the manuscript. Authors should clearly highlight the novelty of the study by presenting novel things in the Introduction section and in other parts of the manuscript. 

Authors have to use the Sensors template to prepare the manuscript according to Journal guidelines!

In the conclusions section main achievements should be higlighted in terms of their possible practical application for cells monitoring.

Why samples were coated by poly-l-lisine?

References list required some minor improvements to meet the journal standards.

Round 2

Reviewer 1 Report

Given the special issue does not require any novelty finding but a description of AFM application, I support the publication.